# Design, Synthesis, Antifungal Activity, and Molecular Docking of Streptochlorin Derivatives Containing the Nitrile Group

**DOI:** 10.3390/md21020103

**Published:** 2023-01-31

**Authors:** Jing-Rui Liu, Ya Gao, Bing Jin, Dale Guo, Fang Deng, Qiang Bian, Hai-Feng Zhang, Xin-Ya Han, Abdallah S. Ali, Ming-Zhi Zhang, Wei-Hua Zhang, Yu-Cheng Gu

**Affiliations:** 1Jiangsu Key Laboratory of Pesticide Science, College of Sciences, Nanjing Agricultural University, Nanjing 210095, China; 2State Key Laboratory Breeding Base of Systematic Research Development and Utilization of Chinese Medicine Resources, School of Pharmacy, Chengdu University of Traditional Chinese Medicine, Chengdu 611137, China; 3National Pesticide Engineering Research Center (Tianjin), College of Chemistry, Nankai University, Tianjin 300071, China; 4Department of Plant Pathology, College of Plant Protection, Nanjing Agricultural University, Nanjing 210095, China; 5School of Chemistry & Chemical Engineering, Anhui University of Technology, Ma’anshan 243002, China; 6Department of Microbiology, Faculty of Agriculture, Cairo University, Giza 12613, Egypt; 7Syngenta Jealott’s Hill International Research Centre, Bracknell RG42 6EY, Berkshire, UK

**Keywords:** streptochlorin, pimprinine, nitrile group, synthesis, antifungal activity, SAR, molecular docking

## Abstract

Based on the structures of natural products streptochlorin and pimprinine derived from marine or soil microorganisms, a series of streptochlorin derivatives containing the nitrile group were designed and synthesized through acylation and oxidative annulation. Evaluation for antifungal activity showed that compound **3a** could be regarded as the most promising candidate—it demonstrated over 85% growth inhibition against *Botrytis cinerea, Gibberella zeae,* and *Colletotrichum lagenarium*, as well as a broad antifungal spectrum in primary screening at the concentration of 50 μg/mL. The SAR study revealed that non-substituent or alkyl substituent at the 2-position of oxazole ring were favorable for antifungal activity, while aryl and monosubstituted aryl were detrimental to activity. Molecular docking models indicated that **3a** formed hydrogen bonds and hydrophobic interactions with Leucyl-tRNA Synthetase, offering a perspective for the possible mechanism of action for antifungal activity of the target compounds.

## 1. Introduction

Natural products are well known as one of the most important sources for lead discovery in medicinal and agricultural chemistry, because their novel scaffolds can afford an opportunity to discover novel candidates with different modes of action from the existing agents. Streptochlorin is a marine natural product with the structure of 4-chloro-5-(3-indolyl)oxazole; it has been reported to display a range of biological activity [1,2,3,4]. Pimprinine is an indole alkaloid produced by many species of *Streptomyces*, first isolated from the filtrates of *Streptomyces pimprina* cultures in 1963 [5,6]; it is a monoamine oxidase (MAO) inhibitor. Both of these natural products belong to the class of naturally occurring 5-(3′-indolyl)oxazoles, and compounds of this family, including Pimprinethine; Pimprinaphine; WS-30581 A and B; Labradorins 1 and 2; Almazole A, B, and C; and Martefragin A, exhibit a wide range of potent biological activities [7] (Figure 1), such as anti-angiogenesis [3], antibiotic [8], anticancer [1], anti-cell proliferation [9], antioxidant [10], and antiviral activity [11,12]. Bioassay conducted at Syngenta showed that streptochlorin and pimprinine are also promising antifungal substances demonstrating good bioactivity against many phytopathogens [13,14,15,16]; for example, streptochlorin displayed excellent antifungal activity against *Pythium dissimile*, *Botrytis cinerea*, *Zymoseptoria tritici*, *Pyriculariaory zae*, *Fusarium culmorum,* and *Rhizoctonia solani* in artificial media. Meanwhile, these compounds lack potency at lower concentrations, rarely warranting further study.

In classical medicinal chemistry, the nitrile group was commonly considered as bioisosteres of carbonyl, hydroxyl, and carboxyl groups, as well as halogen atoms [17]. As nitrile-containing drugs account for 2.4% of the 2327 approved small-molecule drugs according to the DrugBank database by 2018 [18], the presence of the nitrile group in the structure of compounds is a very common feature of drug molecules [19,20], such as Enzalutamide, a hormone treatment that blocks testosterone from reaching prostate cancer cells [21], Escitalopram, a medication used in the management and treatment of major depressive disorder and generalized anxiety disorder [22]; Tofacitinib, as an oral JAK3 inhibitor to treat adults with moderately to severely active rheumatoid arthritis [23]; Verapamil, a medication for treating hypertension, angina, and certain heart rhythm disorders [24]; Rilpivirine, a non-nucleoside reverse transcriptase inhibitor that inhibits the replication of HIV-1 [25]; and Vildagliptin, an orally administered dipeptidyl peptidase-4 (DPP-4) inhibitor for treating diabetes [26]. Meanwhile, Cyazofamid is a novel fungicide exhibiting specific activity against diseases caused by Oomycetes [27]; Azoxystrobin is a broad-spectrum β-methoxyacrylate fungicide that was first introduced in 1998, which inhibits mitochondrial respiration by binding to the Qo site of the cytochrome *bc*_1_ complex [28,29]; and Phenamacril is a Fusarium-specific fungicide used for *Fusarium* head blight management [30,31]. (Figure 2).

Introducing the nitrile group into the molecules is an effective protocol for structural optimization (Figure 3). For example, the nitrile-containing structure exhibited a 277-fold improvement in potency over the non-substituted structure as selective inhibitors of cFMS. For casein kinase 2 (CK2) inhibitor, the nitrile-containing structure improved binding affinity more than 90-fold compared with the non-substituted structure, and the nitrile group was engaged in hydrogen bond interactions with the conserved water molecules in a cocrystal structure (PDB Code: 5H8B) [17].

In this study, based on the parent structures of streptochlorin and pimprinine (Figure 4), we designed and synthesized a series of streptochlorin derivatives containing the nitrile group, and carried out the evaluation for antifungal activity, aiming at the discovery of natural product derivatives with improved antifungal activity. Furthermore, the structure–activity relationships (SARs) around these compounds and the molecular docking of the most active compound with potential target enzyme were further performed.

## 2. Results and Discussion

### 2.1. Synthetic Chemistry

The series of streptochlorin derivatives containing the nitrile group were synthesized as shown in Figure 1, using the reported methods [32,33]. The synthesis started with cheap and readily available indole (**1**). After the acylation of indole, 3-cyanoacetylindole (**2**) was obtained. Then, the target compounds **3** were synthesized by the oxidative annulation of 3-cyanoacetylindole. With DMF as solvent and TBHP as oxidant, 3-cyanoacetylindole reacted with methylene amine under the catalysis of iodine to give compounds **3**. The structures and yields of 20 target compounds are shown in Figure 5. Copies of the NMR spectra and HR-MS (ESI) spectra can be found in the Appendix A.

### 2.2. Antifungal Activity and Structure–Activity Relationships (SARs)

The antifungal activity of the target compounds and positive controls was evaluated against six common phytopathogenic fungi at the concentration of 50 μg/mL, including *Botrytis cinerea* (BOT), *Alternaria solani* (ALS), *Gibberella zeae* (GIB), *Rhizoctorzia solani* (RHI), *Colletotrichum lagenarium* (COL), and *Alternaria Leaf Spot* (ALL). The screening results are presented in Table 1.

As compounds **3a**, **3b**, **3g**, and **3h** exhibited relatively good antifungal activity in primary screening; EC_50_ values of them and commercial fungicides Boscalid and Carbendazim were further determined (Table 2). The most active compound **3a** was compared with Osthole, Boscalid, and Flutriafol in the radar chart shown in Figure 6, and its antifungal activity against four kinds of fungi was more active than at least one of the positive controls.

Although the antifungal activity of most of streptochlorin derivatives containing the nitrile group was relatively poor, making it difficult to find clear structure–activity relationships, some preliminary conclusions could still be drawn.

Firstly, it is worth noting that the target compounds lack antifungal activity potency, though compounds **3a** and **3g** showed a more than 50% antifungal effect against at least three kinds of fungi. **3a** could be regarded as the most promising candidate, as it demonstrated over 85% growth inhibition against *Botrytis cinerea, Gibberella zeae,* and *Colletotrichum lagenarium*, as well as a broad antifungal spectrum.

Secondly, this series of streptochlorin derivatives showed relatively stronger antifungal activity against *Rhizoctorzia solani* than the other five phytopathogenic fungi. This was highlighted by compounds **3b**, **3g**, and **3h**, which were equivalent to or even more active than Osthole.

Thirdly, the antifungal activity data indicated that non-substituent or alkyl substituent at the 2-position of oxazole ring were favorable for antifungal activity, while aryl and monosubstituted aryl were detrimental to activity, though compound **3h** also demonstrated 67.5% growth inhibition against *Rhizoctorzia solani*. This might be due to the presence of methylene on the benzyl group.

### 2.3. Molecular Docking

Although streptochlorin and pimprinine exhibited widely potent biological activities, the mechanism of action for the antifungal activity is still unclear. In our previous studies [16,34], molecular docking was performed on streptochlorin, which indicated that streptochlorin binds with *t*LeuRS in a similar mode to AN2690, and provided some ideas for the possible mechanism of action for antifungal activity of synthesized target compounds.

Molecular docking of the most active compound **3a** with receptor protein *t*LeuRS (PDB Code: 2V0C) was performed using Autodock 4.2. The protein was downloaded in high resolution solved at 1.85 Å from RCSB Protein Data Bank (https://www.rcsb.org/, accessed on 29 October 2022). After the molecular docking, the best binding mode of **3a** (cyan in Figure 7) was selected and analyzed according to the minimum value of the docking energy.

The simulated binding models indicated that compound **3a** formed hydrogen bonds and hydrophobic interactions with the amino acid residues. The nitrile group of **3a** formed a hydrogen bond with residue Met338, the indole N-H bond formed hydrogen bonds with Thr247 and Thr252, and the oxazole ring formed a weak hydrogen bond with Arg346. The indole ring formed hydrophobic interactions with Arg249, Thr252, Val340, His343, and Asp344 (Figure 7).

## 3. Materials and Methods

### 3.1. Chemicals

All commercially available chemicals were purchased from Nanjing Crystal Chemical Co., Ltd. (Nanjing, China) or Alfa Aesar (Beijing, China) and were analytically pure. The specification of silica gel for column chromatography was 200–300 mesh. All target compounds were characterized by melting point, ^1^H NMR, ^13^C NMR, and HR-MS (ESI). The instruments were Büchi M-560 melting point apparatus, Bruker Avance 400 MHz spectrometer (Rheinstetten, Germany), Agilent Technologies 6540 UHD Q-TOF LC-MS (Palo Alto, CA, USA).

Furthermore, 3-cyanoacetylindole (**2**) and the target compounds (**3**) were synthesized using the reported methods [32,33]. All of the reaction yields were not optimized.

#### 3.1.1. Preparation of 3-(1*H*-indol-3-yl)-3-oxopropanenitrile (**2**)

Cyanoacetic acid (3.40 g, 40 mmol) was dissolved in Ac_2_O (76 mL) with stirring and heating to 50 °C. Indole (4.69 g, 40 mmol) was then added and the solution was heated to 85 °C. The reaction was monitored by TLC and, after the reaction was complete, the mixture was cooled in ice water. The solid was collected under suction and washed with MeOH to obtain pure compound **2**.

3-(1*H*-indol-3-yl)-3-oxopropanenitrile (**2**): Orange solid, yield: 66%. ^1^H NMR (400 MHz, DMSO-*d_6_*) δ 12.20 (s, 1H), 8.39 (d, *J* = 3.2 Hz, 1H), 8.17–8.14 (m, 1H), 7.53–7.50 (m, 1H), 7.29–7.21 (m, 2H), 4.51 (s, 2H).

#### 3.1.2. General Procedure for the Synthesis of 2-substituted-4-cyano-5-(1*H*-indol-3-yl)oxazole (**3**)

Compound **2** (0.2 g, 1.5 mmol), amine (1.5 mmol), I_2_ (0.095 g, 0.375 mmol), and TBHP (0.58 mL, 6 mmol) were dissolved in DMF (10mL) and reacted at 60 °C for 6 h. Then, the solvent was concentrated under reduced pressure. Then, CH_2_Cl_2_ was added to the mixture and washed with 50 mL water and 30 mL saturated brine solution. The organic layer was dried over anhydrous Na_2_SO_4_ and the solution was removed under reduced pressure. Finally, the pure product **3** was obtained after purification by column chromatography on silica gel (eluent: petroleum ether/ethyl acetate = 8:1).

### 3.2. Compound Data

#### 3.2.1. 5-(1*H*-indol-3-yl)oxazole-4-carbonitrile (**3a**)

Yellow solid, yield: 46%, m.p. 167.5–168.8 °C. ^1^H NMR (400 MHz, DMSO-*d_6_*) δ 12.16 (s, 1H), 8.65 (s, 1H), 8.14(d, *J* = 3.2 Hz, 1H), 7.98 (dd, *J* = 7.6, 3.6 Hz, 1H), 7.61–7.56 (m, 1H), 7.33–7.22 (m, 2H).^13^C NMR (100 MHz, DMSO-*d_6_*) δ 156.9, 150.8, 136.3, 127.5, 123.6, 123.2, 121.5, 119.9, 114.8, 112.8, 102.8, 101.0. HR-MS (ESI): *m/z* calcd for C_12_H_7_N_3_O ([M + H]^+^) 210.0662, Found 210.0656.

#### 3.2.2. 5-(1*H*-indol-3-yl)-2-methyloxazole-4-carbonitrile (**3b**)

Yellow solid, yield: 85%, m.p. 226.7–227.9 °C. ^1^H NMR (400 MHz, DMSO-*d_6_*) δ 12.08 (s, 1H), 8.06 (d, *J* = 2.8 Hz, 1H), 7.98 (d, *J* = 7.6 Hz, 1H), 7.56 (d, *J* = 8.0 Hz, 1H), 7.25 (ddd, *J* = 15.2, 13.6, 6.8 Hz, 2H), 2.56 (s, 3H). ^13^C NMR (100 MHz, DMSO-*d_6_*) δ 159.8, 156.9, 136.3, 126.9, 123.6, 123.1, 121.3, 120.0, 114.9, 112.7, 102.9, 101.1, 13.5. HR-MS (ESI): *m/z* calcd for C_13_H_9_N_3_O ([M + H]^+^) 224.0818, Found 224.0821.

#### 3.2.3. 2-ethyl-5-(1*H*-indol-3-yl)oxazole-4-carbonitrile (**3c**)

Yellow solid, yield: 40%, m.p. 165.2–167.0 °C. ^1^H NMR (400 MHz, DMSO-*d_6_*) δ 12.09 (s, 1H), 8.07 (d, *J* = 2.8 Hz, 1H), 7.97 (d, *J* = 7.6 Hz, 1H), 7.56 (d, *J* = 7.6 Hz, 1H), 7.31–7.20 (m, 2H), 2.91 (q, *J* = 7.6 Hz, 2H), 1.33 (t, *J* = 7.6 Hz, 3H). ^13^C NMR (100 MHz, DMSO-*d_6_*) δ 163.8, 156.7, 136.3, 126.9, 123.6, 123.1, 121.3, 119.9, 115.0, 112.7, 102.8, 101.2, 20.9, 10.6. HR-MS (ESI): *m/z* calcd for C_14_H_11_N_3_O ([M + H]^+^) 238.0975, Found 238.0983.

#### 3.2.4. 5-(1*H*-indol-3-yl)-2-propyloxazole-4-carbonitrile (**3d**)

Yellow solid, yield: 25%, m.p. 167.3–168.9 °C. ^1^H NMR (400 MHz, DMSO-*d_6_*) δ 12.09 (s, 1H), 8.07 (d, *J* = 2.8 Hz, 1H), 7.97 (d, *J* = 7.6 Hz, 1H), 7.56 (d, *J* = 8.0 Hz, 1H), 7.26 (ddd, *J* = 15.2, 13.6, 6.8 Hz, 2H), 2.87 (t, *J* = 7.2 Hz, 2H), 1.80 (h, *J* = 7.2 Hz, 2H), 1.00 (t, *J* = 7.6 Hz, 3H). ^13^C NMR (100 MHz, DMSO-*d_6_*) δ 162.7, 156.8, 136.3, 126.9, 123.6, 123.1, 121.3, 119.9, 115.0, 112.7, 102.8, 101.2, 29.0, 19.7, 13.5. HR-MS (ESI): *m/z* calcd for C_15_H_13_N_3_O ([M + H]^+^) 252.1131, Found 252.1130.

#### 3.2.5. 2-butyl-5-(1*H*-indol-3-yl)oxazole-4-carbonitrile (**3e**)

Yellow solid, yield: 69%, m.p. 150.2–151.0 °C. ^1^H NMR (400 MHz, DMSO-*d_6_*) δ 12.09 (s, 1H), 8.07 (d, *J* = 2.8 Hz, 1H), 7.97 (d, *J* = 7.6 Hz, 1H), 7.56 (d, *J* = 7.6 Hz, 1H), 7.26 (ddd, *J* = 15.2, 13.6, 6.8 Hz, 2H), 2.89 (t, *J* = 7.6 Hz, 2H), 1.81–1.70 (m, 2H), 1.46–1.35 (m, 2H), 0.94 (t, *J* = 7.2 Hz, 3H). ^13^C NMR (100 MHz, DMSO-*d_6_*) δ 162.8, 156.7, 136.3, 126.9, 123.6, 123.0, 121.3, 119.9, 114.9, 112.7, 102.8, 101.2, 28.2, 26.8, 21.6, 13.6. HR-MS (ESI): *m/z* calcd for C_16_H_15_N_3_O ([M + H]^+^) 266.1288, Found 266.1290.

#### 3.2.6. 5-(1*H*-indol-3-yl)-2-pentyloxazole-4-carbonitrile (**3f**)

Yellow solid, yield: 37%, m.p. 126.5–127.2 °C. ^1^H NMR (400 MHz, DMSO-*d_6_*) δ 12.10 (s, 1H), 8.07 (d, *J* = 2.8 Hz, 1H), 7.97 (d, *J* = 7.6 Hz, 1H), 7.57 (d, *J* = 8.0 Hz, 1H), 7.33–7.20 (m, 2H), 2.87 (t, *J* = 7.6 Hz, 2H), 1.83–1.71 (m, 2H), 1.41–1.30 (m, 4H), 0.89 (t, *J* = 7.2 Hz, 3H). ^13^C NMR (100 MHz, DMSO-*d_6_*) δ 162.8, 156.8, 136.3, 126.9, 123.7, 123.1, 121.3, 119.9, 115.0, 112.8, 102.9, 101.3, 30.7, 27.1, 25.8, 21.9, 13.9. HR-MS (ESI): *m/z* calcd for C_17_H_17_N_3_O ([M + H]^+^) 280.1444, Found 280.1444.

#### 3.2.7. 5-(1*H*-indol-3-yl)-2-isobutyloxazole-4-carbonitrile (**3g**)

Yellow solid, yield: 36%, m.p. 125.1–126.5 °C. ^1^H NMR (400 MHz, DMSO-*d_6_*) δ 12.10 (s, 1H), 8.08 (d, *J* = 2.8 Hz, 1H), 7.97 (d, *J* = 7.6 Hz, 1H), 7.57 (d, *J* = 7.6 Hz, 1H), 7.32–7.19 (m, 2H), 2.78 (d, *J* = 7.2 Hz, 2H), 2.18 (dp, *J* = 13.6, 6.8 Hz, 1H), 1.01 (d, *J* = 6.8 Hz, 6H). ^13^C NMR (100 MHz, DMSO-*d_6_*) δ 162.1, 156.8, 136.3, 126.9, 123.6, 123.1, 121.3, 119.8, 114.9, 112.7, 102.8, 101.2, 35.9, 26.9, 22.1. HR-MS (ESI): *m/z* calcd for C_16_H_15_N_3_O ([M + H]^+^) 266.1288, Found 266.1285.

#### 3.2.8. 2-benzyl-5-(1*H*-indol-3-yl) oxazole-4-carbonitrile (**3h**)

Yellow solid, yield: 88%, m.p. 163.3–164.9 °C. ^1^H NMR (400 MHz, DMSO-*d_6_*) δ 12.11 (s, 1H), 8.08 (d, *J* = 2.8 Hz, 1H), 7.90 (d, *J* = 8.0 Hz, 1H), 7.56 (dd, *J* = 8.0, 0.8 Hz, 1H), 7.44–7.37 (m, 4H), 7.33–7.25 (m, 2H), 7.22 (ddd, *J* = 8.0, 7.2, 1.2 Hz, 1H), 4.31(s, 2H). ^13^C NMR (100 MHz, DMSO-*d_6_*) δ 161.3, 157.1, 136.3, 135.0, 129.1, 128.8, 127.2, 127.1, 123.6, 123.1, 121.4, 119.9, 114.8, 112.8, 103.0, 101.0, 59.8, 33.4, 14.2. HR-MS (ESI): *m/z* calcd for C_19_H_13_N_3_O ([M + H]^+^) 300.1131, Found 300.1129.

#### 3.2.9. 5-(1*H*-indol-3-yl)-2-phenyloxazole-4-carbonitrile (**3i**)

Yellow solid, yield: 70%, m.p. 216.7–218.7 °C. ^1^H NMR (400 MHz, DMSO-*d_6_*) δ 12.20 (s, 1H), 8.14 (d, *J* = 33.2 Hz, 4H), 7.61 (s, 4H), 7.32 (d, *J* = 2.8 Hz, 2H). ^13^C NMR (100 MHz, DMSO-*d_6_*) δ 158.5, 157.1, 136.3, 131.4, 129.4, 128.6, 127.8, 127.6, 126.2, 125.5, 123.6, 123.2, 121.6, 120.1, 114.8, 112.8, 104.2, 101.2. HR-MS (ESI): *m/z* calcd for C_18_H_11_N_3_O ([M + H]^+^) 286.0975, Found 286.0974.

#### 3.2.10. 5-(1*H*-indol-3-yl)-2-(o-tolyl) oxazole-4-carbonitrile (**3j**)

Yellow solid, yield: 35%, m.p. 226.0–227.3 °C. ^1^H NMR (400 MHz, DMSO-*d_6_*) δ 12.20 (s, 1H), 8.42–7.95 (m, 3H), 7.71–7.03 (m, 6H), 2.72 (s, 3H). ^13^C NMR (100 MHz, DMSO-*d_6_*) δ 158.6, 156.5, 137.3, 136.3, 131.8, 130.8, 128.5, 127.5, 126.5, 124.4, 123.6, 123.1, 121.5, 119.9, 115.0, 112.8, 104.2, 101.1, 21.8. HR-MS (ESI): *m/z* calcd for C_19_H_13_N_3_O ([M + H]^+^) 300.1131, Found 300.1132.

#### 3.2.11. 2-(2-fluorophenyl)-5-(1*H*-indol-3-yl) oxazole-4-carbonitrile (**3k**)

Yellow solid, yield: 64%, m.p. 221.0–222.5 °C. ^1^H NMR (400 MHz, DMSO-*d_6_*) δ 12.22 (s, 1H), 8.20 (s, 1H), 8.17–8.10 (m, 2H), 7.67 (dd, *J* = 12.8, 6.4 Hz, 1H), 7.60 (d, *J* = 7.2 Hz, 1H), 7.54–7.43 (m, 2H), 7.34–7.27 (m, 2H). ^13^C NMR (100 MHz, DMSO-*d_6_*) δ 160.4, 158.3, 157.2, 154.7 (d, *J* = 5.2 Hz), 136.3, 129.2, 127.4, 125.2 (d, *J* = 3.6 Hz), 123.6, 123.2, 121.5, 119.9, 117.1 (d, *J* = 21.2 Hz), 114.7, 113.7 (d, *J* = 11.2Hz), 112.7, 103.9, 101.1. HR-MS (ESI): *m/z* calcd for C_18_H_10_FN_3_O ([M + H]^+^) 304.0881, Found 304.0882.

#### 3.2.12. 5-(1*H*-indol-3-yl)-2-(m-tolyl) oxazole-4-carbonitrile (**3l**)

Yellow solid, yield: 63%, m.p. 193.2–194.5 °C. ^1^H NMR (400 MHz, DMSO-*d_6_*) δ 12.20 (s, 1H), 8.21 (s, 1H), 8.16–8.10 (m, 1H), 7.93 (s, 2H), 7.63–7.56 (m, 1H), 7.51 (t, *J* = 7.6 Hz, 1H), 7.43 (d, *J* = 7.6 Hz, 1H), 7.34–7.29 (m, 2H), 2.44 (s, 3H). ^13^C NMR (100 MHz, DMSO-*d_6_*) δ 158.4, 157.0, 138.7, 136.3, 131.9, 129.1, 127.4, 126.4, 125.4, 123.6, 123.3, 123.1, 121.5, 120.1, 114.9, 112.7, 104.1, 101.3, 21.0. HR-MS (ESI): *m/z* calcd for C_19_H_13_N_3_O ([M + H]^+^) 300.1131, Found 300.1131.

#### 3.2.13. 5-(1*H*-indol-3-yl)-2-(3-methoxyphenyl) oxazole-4-carbonitrile (**3m**)

Yellow solid, yield: 62%, m.p. 185.9–187.2 °C. ^1^H NMR (400 MHz, DMSO-*d_6_*) δ 12.22 (s, 1H), 8.23 (d, *J* = 2.8 Hz, 1H), 8.14–8.08 (m, 1H), 7.71 (d, *J* = 7.6 Hz, 1H), 7.63–7.51 (m, 3H), 7.36–7.27 (m, 2H), 7.24–7.15 (m, 1H), 3.88 (s, 3H). ^13^C NMR (100 MHz, DMSO-*d_6_*) δ 159.7, 158.2, 157.1, 136.4, 130.7, 127.6, 126.7, 123.6, 123.2, 121.7, 120.0, 118.5, 117.2, 114.9, 112.8, 111.1, 104.1, 101.2, 55.4. HR-MS (ESI): *m/z* calcd for C_19_H_13_N_3_O_2_ ([M + H]^+^) 316.1081, Found 316.1081.

#### 3.2.14. 2-(3-fluorophenyl)-5-(1*H*-indol-3-yl) oxazole-4-carbonitrile (**3n**)

Yellow solid, yield: 72%, m.p. 226.7–227.9 °C. ^1^H NMR (400 MHz, DMSO-*d_6_*) δ 12.23 (s, 1H), 8.23 (s, 1H), 8.14 (d, *J* = 4.8 Hz, 1H), 7.96 (d, *J* = 7.6 Hz, 1H), 7.87 (d, *J* = 9.6 Hz, 1H), 7.68 (dd, *J* = 14.0, 7.6 Hz, 1H), 7.62–7.56 (m, 1H), 7.47 (t, *J* = 7.6 Hz, 1H), 7.35–7.27 (m, 2H). ^13^C NMR (100 MHz, DMSO-*d_6_*) δ 163.6, 161.2, 157.4, 157.2 (d, *J* = 3.2 Hz), 136.3, 131.7 (d, *J* = 8.4 Hz), 127.8, 127.5 (d, *J* = 8.8 Hz), 123.5, 123.2, 122.4, 121.6, 120.1, 118.3 (d, *J* = 21.2 Hz), 114.7, 112.8 (t, *J* = 12.0 Hz), 104.2, 101.0. HR-MS (ESI): *m/z* calcd for C_18_H_10_FN_3_O ([M + H]^+^) 304.0881, Found 304.0888.

#### 3.2.15. 2-(3-bromophenyl)-5-(1*H*-indol-3-yl) oxazole-4-carbonitrile (**3o**)

Yellow solid, yield: 48%, m.p. 244.3–245.9 °C. ^1^H NMR (400 MHz, DMSO-*d_6_*) δ 12.24 (s, 1H), 8.24 (d, *J* = 3.2 Hz, 1H), 8.19 (t, *J* = 1.6 Hz, 1H), 8.13–8.09 (m, 2H), 7.83–7.79 (m, 1H), 7.62–7.55 (m, 2H), 7.35–7.28 (m, 2H). ^13^C NMR (100 MHz, DMSO-*d_6_*) δ 157.4, 156.8, 136.3, 134.0, 131.5, 128.4, 127.9, 127.5, 125.2, 123.5, 123.2, 122.5, 121.6, 120.0, 114.7, 112.8, 104.2, 101.0. HR-MS (ESI): *m/z* calcd for C_18_H_10_BrN_3_O ([M + H]^+^) 364.0080, Found 364.0077.

#### 3.2.16. 5-(1*H*-indol-3-yl)-2-(p-tolyl) oxazole-4-carbonitrile (**3p**)

Yellow solid, yield: 50%, m.p. 245.4–247.6 °C. ^1^H NMR (400 MHz, DMSO-*d_6_*) δ 12.19 (s, 1H), 8.21 (d, *J* = 2.8 Hz, 1H), 8.15–8.10 (m, 1H), 8.01 (d, *J* = 8.0 Hz, 2H), 7.62–7.56 (m, 1H), 7.43 (d, *J* = 8.0 Hz, 2H), 7.31 (p, *J* = 5.6 Hz, 2H), 2.42 (s, 3H). ^13^C NMR (100 MHz, DMSO-*d_6_*) δ 158.6, 156.8, 141.4, 136.3, 129.9, 129.1, 127.8, 127.4, 126.1, 123.6, 123.1, 122.8, 121.6, 120.1, 114.9, 112.8, 104.0, 101.2, 21.2. HR-MS (ESI): *m/z* calcd for C_19_H_13_N_3_O ([M + H]^+^) 300.1131, Found 300.1135.

#### 3.2.17. 2-(4-fluorophenyl)-5-(1*H*-indol-3-yl) oxazole-4-carbonitrile (**3q**)

Yellow solid, yield: 67%, m.p. 255.8–258.0 °C. ^1^H NMR (400 MHz, DMSO-*d_6_*) δ 12.20 (s, 1H), 8.26–8.08 (m, 4H), 7.59 (d, *J* = 6.4 Hz, 1H), 7.46 (t, *J* = 8.0 Hz, 2H), 7.36–7.26 (m, 2H). ^13^C NMR (100 MHz, DMSO- *d_6_*) δ 162.9, 157.7, 157.2, 136.3, 128.9, 128.8, 127.6, 123.6, 123.2, 122.2 (d, *J* = 3.2 Hz), 121.6, 120.1, 116.7, 116.5, 114.8, 112.8, 104.1, 101.1. HR-MS (ESI): *m/z* calcd for C_18_H_10_FN_3_O ([M + H]^+^) 304.0881, Found 304.0874.

#### 3.2.18. 2-(4-chlorophenyl)-5-(1*H*-indol-3-yl) oxazole-4-carbonitrile (**3r**)

Yellow solid, yield: 28%, m.p. 225.6–226.5 °C. ^1^H NMR (400 MHz, DMSO-*d_6_*) δ 12.22 (s, 1H), 8.32–8.00 (m, 4H), 7.64 (dd, *J* = 38.0, 8.4 Hz, 3H), 7.30 (dd, *J* = 9.2, 5.6 Hz, 2H). ^13^C NMR (100 MHz, DMSO-*d_6_*) δ 157.5, 157.3, 136.3, 136.1, 129.5, 127.9, 127.7, 124.3, 123.5, 123.2, 121.6, 120.1, 114.7, 112.8, 104.2, 101.1, 99.6. HR-MS (ESI): *m/z* calcd for C_18_H_10_ClN_3_O ([M + H]^+^) 320.0585, Found 320.0582.

#### 3.2.19. 5-(1*H*-indol-3-yl)-2-(4-(trifluoromethyl)phenyl)oxazole-4-carbonitrile (**3s**)

Yellow solid, yield: 55%, m.p. 274.2–275.9 °C. ^1^H NMR (400 MHz, DMSO-*d_6_*) δ 12.23 (s, 1H), 8.24 (d, *J* = 8.0 Hz, 2H), 8.19 (d, *J* = 3.2 Hz, 1H), 8.12–8.08 (m, 1H), 7.92 (d, *J* = 8.4 Hz, 2H), 7.60–7.56 (m, 1H), 7.33–7.27 (m, 2H). ^13^C NMR (100 MHz, DMSO-*d_6_*) δ 157.8, 157.0, 136.3, 130.8, 129.1, 128.0, 126.9, 126.3, 123.6 (d, *J* = 7.6 Hz), 123.3, 121.7, 120.0 (d, *J* = 11.6 Hz), 114.6, 112.9, 104.4, 101.0 (d, *J* = 6.8 Hz), 99.6. HR-MS (ESI): *m/z* calcd for C_19_H_10_F_3_N_3_O ([M + H]^+^) 354.0849, Found 354.0851.

#### 3.2.20. 5-(1*H*-indol-3-yl)-2-(thiophen-2-yl)oxazole-4-carbonitrile (**3t**)

Yellow solid, yield: 69%, m.p. 207.7–208.9 °C. ^1^H NMR (400 MHz, DMSO-*d_6_*) δ 12.21 (s, 1H), 8.17 (d, *J* = 2.8 Hz, 1H), 8.09 (dd, *J* = 6.8, 1.6 Hz, 1H), 7.92 (ddd, *J* = 6.0, 4.4, 1.2 Hz, 2H), 7.59 (dt, *J* = 7.6, 3.2 Hz, 1H), 7.33–7.28 (m, 3H). ^13^C NMR (100 MHz, DMSO-*d6*) δ 156.6, 154.8, 136.3, 130.9, 129.5, 128.8, 127.7, 127.4, 123.6, 123.2, 121.6, 120.1, 114.6, 112.8, 103.9, 100.9. HR-MS (ESI): *m/z* calcd for C_16_H_9_N_3_OS ([M + H]^+^) 292.0539, Found 292.0538.

### 3.3. Biological Assays

Antifungal activity testing was carried out using the mycelia growth-inhibitory rate method. The six common phytopathogenic fungi selected were *Botrytis cinerea*, *Alternaria solani, Gibberella zeae*, *Rhizoctonia solani*, *Colletotrichum lagenarium*, and *Alternaria leaf spot*, which were provided by the Laboratory of Plant Disease Control, Nanjing Agricultural University. The experimental procedure of the antifungal activity was performed according to the paper from the Department of Plant Pathology, Nanjing Agricultural University [35]. The compounds and three positive controls, Osthole, Boscalid, and Flutriafol, were tested at 50 μg/mL in the primary screening. The strains were activated in Potato Dextrose Agar Medium (PDA) at 25 °C for 2–15 days to afford new mycelia; the edge of the mycelia was punched before the antifungal activity assay. The screening results are listed in Table 1.

### 3.4. Molecular Docking Strategy

First, removing the water molecules in the protein was performed using PyMol 2.5.4 (Schrödinger, New York, NY, USA). Drawing and energy minimization of ligand molecules were completed in Chemdraw (Version 14.0, CambridgeSoft, Cambridge, MA, USA) and Chem3D (Version 14.0, CambridgeSoft, Cambridge, MA, USA). Then, the preparation of the protein and ligand was performed using Autodock 4.2 (The Scripps Research Institute, La Jolla, CA, USA). For protein, we added the hydrogen atoms, calculated the charge, and added the atom type (Assign AD4type). As for ligand, we checked the charge, “detect Root”, and “Choose Torsions”. Finally, we ran docking after setting the Grid (center_x = 53.489, center_y = −26.319, center_z = 33.004, size_x = size_y = size_z = 22.5 Å) and docking parameters, and the number of runs was 50. The best binding mode was analyzed in PyMol.

## 4. Conclusions

Based on the natural product structures of streptochlorin and pimprinine derived from marine or soil microorganisms, 20 kinds of streptochlorin derivatives containing the nitrile group were effectively synthesized from indole, through acylation and oxidative annulation. The antifungal activity of the target compounds against six common phytopathogenic fungi was evaluated at 50 μg/mL. Evaluation of antifungal activity showed that compound **3a** could be regarded as the most promising candidate—it demonstrated over 85% growth inhibition against *Botrytis cinerea, Gibberella zeae,* and *Colletotrichum lagenarium*, as well as a broad antifungal spectrum in the primary screening at a concentration of 50 μg/mL, though the target compounds lack antifungal activity potency as a whole. The SAR study revealed that non-substituent or alkyl substituent at the 2-position of oxazole ring were favorable for antifungal activity, while aryl and monosubstituted aryl were detrimental to activity. Molecular docking models indicated that **3a** formed hydrogen bonds and hydrophobic interactions with Leucyl-tRNA Synthetase, offering a perspective for the possible mechanism of action for antifungal activity of the target compounds. Further structural optimization is well under way.

## Data Availability

The data presented in this study are available in the manuscript and in the Appendix A.

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
