# Peer review of "Design, Synthesis, Antifungal Activity, and Molecular Docking of Streptochlorin Derivatives Containing the Nitrile Group"

_marinedrugs, 2023, doi:10.3390/md21020103_

Round 1

Reviewer 1 Report

The 20 novel streptochlorin derivatives, synthesized by the authors, have been satisfactorily characterized by NMR spectroscopy and high resolution mass spectrometry. The spectra in the Supplementary Materials file are accurate and of good quality. The work concerning biological evaluation and docking experiments seems sound. There are quite a few English grammar mistakes but otherwise the English language is overall good. 

After English language corrections the manuscript is suitable for publication in Marine Drugs

Author Response

Manuscript ID: marinedrugs-2176951Title: Design, synthesis, antifungal activity and molecular docking of streptochlorin derivative containing nitrile group  Dear Reviewer, 

Thanks a lot for your kind help, we also appreciate for the positive comments, patient checking and constructive suggestions to improve our manuscript, and this will also encourage us to carry out better research. Based on the comments and suggestions, we have revised our manuscript as described below.

The comments and suggestions:

The 20 novel streptochlorin derivatives, synthesized by the authors, have been satisfactorily characterized by NMR spectroscopy and high resolution mass spectrometry. The spectra in the Supplementary Materials file are accurate and of good quality. The work concerning biological evaluation and docking experiments seems sound. There are quite a few English grammar mistakes but otherwise the English language is overall good. 

After English language corrections the manuscript is suitable for publication in Marine Drugs.

Answer: We appreciate the reviewer for positive comments, which will motivate us to do better research, and also thanks for the patient checking. We have carefully corrected all the grammar and syntax errors we had found. We also checked the explanation of data in the manuscript.

Thanks again for your kind help.

Sincerely yours,

Assoc. Prof. Ming-Zhi Zhang

Nanjing Agricultural University, Nanjing 210095, China

Tel: 86-25-8439-9210

Reviewer 2 Report

The ms was well organized and written, however it still needed to improve.

1. The compound name, such like 3a should be in bold in Table 1 and 2, line 151; The six fungi name should be in italic in labeling of Table 1.

2. The design and synthesis of derivatives was too simple. Based on the research basis of the author, the design should be either on LBDD or on SBDD.

3. The inhibit activity of compounds on Leucyl-tRNA synthetase (kinase?) in cell or cell-free system should be added.

4. Why the Boscalid was selected as positive control, but not the Osthole and Flutriafol?

I may recommend the manuscript for publication after more experiments were performed and carefully revised.

Author Response

Manuscript ID: marinedrugs-2176951

Title: Design, synthesis, antifungal activity and molecular docking of streptochlorin derivative containing nitrile group  Dear Reviewer, 

Thanks a lot for your kind help, we also appreciate for the positive comments, patient checking and constructive suggestions to improve our manuscript, and this will also encourage us to carry out better research. Based on the comments and suggestions, we have revised our manuscript as described below.

The comments and suggestions:

The ms was well organized and written, however it still needed to improve.

  1. The compound name, such like 3ashould be in bold in Table 1 and 2, line 151; The six fungi name should be in italic in labeling of Table 1.

Answer: We appreciate the reviewer for patient checking, and we have carefully corrected all font formats in the manuscript. We also corrected all the grammar and syntax errors we had found.

  1. The design and synthesis of derivatives was too simple. Based on the research basis of the author, the design should be either on LBDD or on SBDD.

Answer: We appreciate for your useful comments. The simple but effective design is our pursuit, and the design of this series of derivatives was based on the parent structures of natural products streptochlorin and pimprinine, as well as the nitrile group. In the early stage, we mainly focused on the structural optimization of compounds, with the aim to discover active compound for the study of action mode. We will also conduct further research on LBDD and SBDD in the next study.

  1. The inhibit activity of compounds on Leucyl-tRNA synthetase (kinase?) in cell or cell-free system should be added.

Answer: Thanks for your kind suggestion. Our plan is to finish the determination of related enzymes activity in the next study, we are now studying the relevant literatures.

  1. Why the Boscalid was selected as positive control, but not the Osthole and Flutriafol?

I may recommend the manuscript for publication after more experiments were performed and carefully revised.

Answer: We are thankful for your positive comments. In primary screenings, Osthole, Boscalid and Flutriafol were used as the positive controls, and Boscalid showed quite different growth inhibition against six selected phytopathogenic fungi. So, we selected Boscalid as the positive control in EC50 determination, which could be more realistic and objective to reflect the activity of our compounds.

Thanks again for your kind help.

Sincerely yours,

Assoc. Prof. Ming-Zhi Zhang

Nanjing Agricultural University, Nanjing 210095, China

Tel: 86-25-8439-9210

Reviewer 3 Report

In the manuscript, on the basis of natural product structures of streptochlorin and pimprinine, 20 streptochlorin derivatives containing the nitrile group were effectively synthesized from indole through acylation with cyanoacetic acid and oxidative annulation with primary amines. Their antifungal activities against six common phytopathogenic fungi were evaluated at the concentration of 50 μg/mL. Biological assays showed that compound 3a was the most promising candidate as it demonstrated over 85% growth inhibition against Botrytis cinerea, Gibberella zeae and Colletotrichum lagenarium, and a broad antifungal spectrum in primary screening. The SAR study revealed the relationship between structure and antifungal activity and provided some information for the development of the new target molecules. The manuscript was well organized and could be accepted for publication after a major revision.

Comments

1.     Compounds 3f, 3h, 3i, 3k, 3m, 3p, 3r, 3s, 3t are not pure enough for publication and they should be further purified before acceptance.

2.     For 13C NMR data, please keep only one digital after point if they are separable. Please delete (s) in 13C NMR data.

3.     English should be improved, such as (not list all errors)

nitrile groups were commonly considered as  -  the nitrile group was commonly considered as

the presence of nitrile groups  -  the presence of the nitrile group

the H-substituted structure  -  the nonsubstituted structure

nitrile group   --  the nitrile group

compound 3a and 3g showed   -   compounds 3a and 3g showed

Author Response

Manuscript ID: marinedrugs-2176951

Title: Design, synthesis, antifungal activity and molecular docking of streptochlorin derivative containing nitrile group  Dear Reviewer, 

Thanks a lot for your kind help, we also appreciate for the positive comments, patient checking and constructive suggestions to improve our manuscript, and this will also encourage us to carry out better research. Based on the comments and suggestions, we have revised our manuscript as described below.

The comments and suggestions:

In the manuscript, on the basis of natural product structures of streptochlorin and pimprinine, 20 streptochlorin derivatives containing the nitrile group were effectively synthesized from indole through acylation with cyanoacetic acid and oxidative annulation with primary amines. Their antifungal activities against six common phytopathogenic fungi were evaluated at the concentration of 50 μg/mL. Biological assays showed that compound 3a was the most promising candidate as it demonstrated over 85% growth inhibition against Botrytis cinerea, Gibberella zeae and Colletotrichum lagenarium, and a broad antifungal spectrum in primary screening. The SAR study revealed the relationship between structure and antifungal activity and provided some information for the development of the new target molecules. The manuscript was well organized and could be accepted for publication after a major revision.

 Comments

  1. Compounds 3f, 3h, 3i, 3k, 3m, 3p, 3r, 3s, 3t are not pure enough for publication and they should be further purified before acceptance.

Answer: We appreciate the reviewer for patient checking. We have been doing research on organic synthesis for a long time and our recent compounds have also been confirmed by NMR, HR-MS, and even single crystal X-ray diffraction. We speculate that some compounds are not pure enough because of a small amount of solvent. Such δ 0.8-0.9, 1.2~1.3 are petroleum ether, and δ 1.17, 1.99, 4.03 are ethyl acetate. δ 2.50 is DMSO-d6. δ 3.33 is water. And for the reason of the Chinese New Year holiday, we are so sorry that we could not further purify the compounds at present, and we’ll update the NMR data of purified target compounds after the holiday. 

  1. For 13C NMR data, please keep only one digital after point if they are separable. Please delete (s) in 13C NMR data.

Answer: We appreciate the reviewer for positive comments and kind help. We have corrected the 13C NMR data.

  1. English should be improved, such as (not list all errors)

nitrile groups were commonly considered as  -  the nitrile group was commonly considered as

the presence of nitrile groups  -  the presence of the nitrile group

the H-substituted structure  -  the nonsubstituted structure

nitrile group   --  the nitrile group

compound 3a and 3g showed   -   compounds 3a and 3g showed

Answer: Thank the reviewer for patient checking. We have carefully corrected all the grammar and syntax errors we had found. We also checked the explanation of data in the manuscript.

Thanks again for your kind help.

Sincerely yours,

Assoc. Prof. Ming-Zhi Zhang

Nanjing Agricultural University, Nanjing 210095, China

Tel: 86-25-8439-9210

Reviewer 4 Report

Authors synthesized streptochlorin derivatives, their antifungal activities and docking studies. 

Suggestions:

1. First of all authors have to eliminate the plagiarism which is alarming (45%), plagiarism from their own publications. It indicates that authors are reproducing the data.

2. Why authors select the protein PDB: 2V0C?  Please indicate your protein selection criteria.

3. The authors have not mention the procedure of docking under heading material and methods. Authors have to describe the methodology and more about the software used.

4. Are the authors dock against the whole protein or a pocket, please indicate?

5. Authors have to put the spectroscopic data in the main manuscript instead of supplementary material.

6. Authors provide binding pattern for only compound 3a, provide the binding pattern and binding energy score for all the compounds. Moreover, I suggest do MD simulations for 50 ns for all the compounds. 

Author Response

Manuscript ID: marinedrugs-2176951

Title: Design, synthesis, antifungal activity and molecular docking of streptochlorin derivative containing nitrile group  

Dear Reviewer, 

Thanks a lot for your kind help, we also appreciate for the positive comments, patient checking and constructive suggestions to improve our manuscript, and this will also encourage us to carry out better research. Based on the comments and suggestions, we have revised our manuscript as described below.

The comments and suggestions:

Authors synthesized streptochlorin derivatives, their antifungal activities and docking studies. 

Suggestions:

  1. First of all authors have to eliminate the plagiarism which is alarming (45%), plagiarism from their own publications. It indicates that authors are reproducing the data.

Answer: We appreciate the reviewer for the patient checking and kind suggestions. Because this is a series of studies, some of the methods and reagents are the same as our previous study. Anyways, we have carefully revised the description in the manuscript, and the repetition rate has been greatly reduced.

  1. Why authors select the protein PDB: 2V0C?  Please indicate your protein selection criteria.

Answer: We are thankful for your positive comments, and we are glad to tell the selection criteria. In our previous study, we found some reported indole alkaloids and analogues, with similar or modified chemical structures of streptochlorin, such as indolmycin, indolmycin derivative and chuangxinmycin, they are potent and selective inhibitors of aminoacyl-tRNA Synthetase (aaRS), which is an important enzyme family for the discovery of antifungal drugs, in which Tavaborole (AN2690), a novel boron-containing small molecule medication for the treatment of fungal infection launched by Anacor Pharmaceuticals in 2014, it inhibits fungal protein synthesis by the inhibition of Leucyl-tRNA Synthetase (LeuRS), a proofreading aaRS. So, we speculate that streptochlorin might also act on the same protein, and streptochlorin is the lead structure of this series of compounds in the manuscript. Therefore, we performed the molecular docking with Leucyl-tRNA Synthetase, and its PDB Code is 2V0C.

  1. The authors have not mention the procedure of docking under heading material and methods. Authors have to describe the methodology and more about the software used.

Answer: We appreciate the reviewer for positive comments and kind suggestions. We are glad to add some details. First, we deleted the water molecules in the protein, which was preformed using PyMol. Drawing and energy minimization of ligand molecule were completed in Chemdraw and Chem3D. Then the preparation of the protein and ligand was performed using Autodock 4.2. For protein, we added the hydrogen atoms, calculated the charge, and added the atom type (Assign AD4type). As for ligand, we checked the charge, “detect Root”, and “Choose Torsions”. Finally, we ran docking after setting the Grid and docking parameters, and the number of runs was 50. The best binding mode was analyzed in PyMol.

  1. Are the authors dock against the whole protein or a pocket, please indicate?

Answer: Thank the reviewer for the question. We chose the pocket in the docking. And the coordinate and size are as follows: 

center_x = 53.489, center_y = -26.319, center_z = 33.004

size_x = 22.5, size_y = 22.5, size_z = 22.5

  1. Authors have to put the spectroscopic data in the main manuscript instead of supplementary material.

Answer: We appreciate for your useful comments. We have transferred the compounds data into the main manuscript.

  1. Authors provide binding pattern for only compound 3a, provide the binding pattern and binding energy score for all the compounds. Moreover, I suggest do MD simulations for 50 ns for all the compounds.

Answer: Thanks for your kind suggestion. As shown in the manuscript, compound 3a showed the best antifungal activity, so 3a was chose as the ligand. We think that the model of other compounds may not show the readers better results. We are also considering do MD simulations for 50 ns for the compounds in the next study.

Thanks again for your kind help.

Sincerely yours,

Assoc. Prof. Ming-Zhi Zhang

Nanjing Agricultural University, Nanjing 210095, China

Tel: 86-25-8439-9210

Round 2

Reviewer 3 Report

The referee still hope to publish pure NMR spectra.  It can be accepted for publication after pure NMR spectra are uploaded.

Author Response

Answer: We appreciate the reviewer's rigorous academic attitude very much, which is very important for cultivating students' realistic and pragmatic spirit, which is also what we have been pursuing. We are very sorry that Chinese universities are currently on Spring Festival holiday, and epidemic control has just been lifted, so we cannot re-purify the compound and complete the NMR sampling in a short time, though we speculate that some compounds are not pure enough because of a small amount of solvent. Such δ 0.8-0.9, 1.2~1.3 are petroleum ether, and δ 1.17, 1.99, 4.03 are ethyl acetate. δ 2.50 is DMSO-d6. δ 3.33 is water. We promise that after returning to campus, we will further purify the compounds mentioned by our reviewer and complete NMR characterization.

Reviewer 4 Report

The authors made changes and the corresponding manuscript has been sufficiently improved. As they are mentioning, they will do molecular dynamics to explore the binding affinity between compounds. 

Author Response

Many thanks to the reviewers for their positive comments.